# Expression Levels of the Immune-Related p38 Mitogen-Activated Protein Kinase Transcript in Response to Environmental Pollutants on *Macrophthalmus japonicus* Crab

**DOI:** 10.3390/genes11090958

**Published:** 2020-08-19

**Authors:** Kiyun Park, Won-Seok Kim, Bohyung Choi, Ihn-Sil Kwak

**Affiliations:** 1Fisheries Science Institute, Chonnam National University, Yeosu 59626, Korea; ecoblue@hotmail.com (K.P.); ecoblue8@gmail.com (B.C.); 2Department of Ocean Integrated Science, Chonnam National University, Yeosu 550-749, Korea; ecoblue7@naver.com

**Keywords:** crustacean, p38 mitogen-activated protein kinase (*p38 MAPK*), perfluorooctane sulfonate (PFOS), di(2-ethylhexyl) phthalate (DEHP), bisphenol A (BPA), irgarol

## Abstract

Environmental pollution in the aquatic environment poses a threat to the immune system of benthic organisms. The *Macrophthalmus japonicus* crab, which inhabits tidal flat sediments, is a marine invertebrate that provides nutrient and organic matter cycling as a means of purification. Here, we characterized the *M. japonicus* p38 mitogen-activated protein kinase (*MAPK*) gene, which plays key roles in the regulation of cellular immune and apoptosis responses. *M. japonicus*
*p38 MAPK* displayed the characteristics of the conserved *MAPK* family with Thr-Gly-Tyr (TGY) motif and substrate-binding site Ala-Thr-Arg-Trp (ATRW). The amino acid sequence of the *M. japonicus p38 MAPK* showed a close phylogenetic relationship to *Eriocheir sinensis MAPK14* and *Scylla paramamosain*
*p38 MAPK*. The phylogenetic tree displayed two origins of *p38 MAPK*: crustacean and insect. The tissue distribution patterns showed the highest expression in the gills and hepatopancreas of *M. japonicus* crab. In addition, *p38 MAPK* expression in *M. japonicus* gills and hepatopancreas was evaluated after exposure to environmental pollutants such as perfluorooctane sulfonate (PFOS), irgarol, di(2-ethylhexyl) phthalate (DEHP), and bisphenol A (BPA). In the gills, *p38 MAPK* expression significantly increased after exposure to all concentrations of the chemicals on day 7. However, on day 1, there were increased *p38 MAPK* responses observed after PFOS and irgarol exposure, whereas decreased *p38 MAPK* responses were observed after DEHP and BPA exposure. The upregulation of *p38 MAPK* gene also significantly led to *M. japonicus* hepatopancreas being undertested in all environmental pollutants. The findings in this study supported that anti-stress responses against exposure to environmental pollutants were reflected in changes in expression levels in *M. japonicus*
*p38 MAPK* signaling regulation as a cellular defense mechanism.

## 1. Introduction

Environmental changes due to anthropogenic pollution continuously cause potential threats to the health of marine organisms and these changes can lead to the accumulation of chemical toxicity in the benthos and sediment ecosystem. Perfluorooctane sulfonate (PFOS) is an increasingly critical environmental pollutant due to its persistence and long-term mobility [1,2]. PFOS was frequently detected in sediment and fish because of the accumulation of PFOS in biota [2]. Irgarol (2-methylthio-4-tert-butylamino-6-cyclopropylamino-striazine), an algaecidal additive in antifouling paints, has been broadly detected in freshwater, seawater, and sediment due to leaching from painted ships and transport [3,4]. Irgarol was detected at concentrations of 2.02 μg L^−^^1^ in coastal waters of Malaysia, and 1.7 μg L^−^^1^ in the surface waters of marinas on the Côte d’ Azur, France [5,6]. High concentrations of irgarol (<1.0–89.7 μg kg^−1^) were observed in sediments from the Brazilian Northeast [7]. Di-(2-ethylhexyl) phthalate (DEHP), a plasticizer, is widely used in plastic products including food packaging, paint, toys, and medical devices [8]. DEHP is one of the most ubiquitous environmental pollutants in drinking water and aquatic resources [9]. In China, the DEHP level was detected to be 45.73 μg g^−1^ in the Yellow River and 1.56 μg g^−1^ as dry weight in the Qiantang river sediment [10]. DEHP was the contaminant quantified with higher concentrations reaching 2.3 μg L^−1^ in reservoirs and distribution tanks around Mexico City [9]. DEHP was also detected at high concentrations of up to 1009.7 μg L^−1^ in a Bogotá reservoir in Colombia [11]. Bisphenol A (BPA), an endocrine-disrupting chemical, is used in the production of epoxy resins and polycarbonate plastic [12]. BPA was detected at a concentration of 4.42 μg L^−1^ in surface water and 1.11 μg L^−1^ in groundwater in Brazil [13,14]. Emerging contaminants have attracted much attention regarding their occurrence in the environment and their potential toxicity to organisms inhabiting the environment in recent years.

Mitogen-activated protein kinases (*MAPKs*) are a class of serine/threonine protein kinases that have a critical function in the cellular response to extracellular stress [15,16]. *p38 MAPK*, a major group of the *MAPK* family, plays an important role in various signal processes including the inflammatory response, cell differentiation, cell cycle regulation, and apoptosis [17,18,19,20]. Increasing studies have reported that activations of *p38 MAPK* can be triggered by various extracellular stressors, including viral infections, environmental stress, and UV irradiation [16,21,22,23,24]. These results indicate that *p38 MAPK* plays a pivotal role in the defense process against cellular stressors.

Mud crabs (*Macrophthalmus japonicus*) inhabit intertidal mudflats throughout the Indo-Pacific estuaries [25,26]. *M. japonicus* is morphologically classified in the order Decapoda and the family Macrophthalmidae. Burrowing behavior by crabs has an important function in oxygen supplement and organic carbon cycling in the coastal sediment environment [27,28]. The *M. japonicus* crab as a bioturbator affects the sedimentary environment and density of other benthic animals through its sediment reworking process and associated alteration in environmental chemical properties [26]. In addition, *M. japonicus* can be used for the characterization and purification of lectin as well as resources of fishing and fishery [26,29]. The intertidal mud crab is a good indicator species for risk assessment of the coastal benthic environment. In crustaceans, the response of *p38 MAPK* signaling to environmental pollutants is limited, although there is some information on the *p38 MAPK* pathway in response to pathogenic infection [15,30,31]. In this study, the *p38 MAPK* gene was sequenced from *M. japonicus*, and its role in response to environmental stressors, such as PFOS, irgarol, DEHP, and BPA, were investigated via the gills and hepatopancreas of *M. japonicus*. The results suggested potential activation of *M. japonicus p38 MAPKs* as a defense immune response against exposure toxicities to environmental pollutants.

## 2. Materials and Methods

### 2.1. Organisms and Pollutants Exposure

Healthy *M. japonicus* mud crabs (body weight: 7 g ± 3 g) were purchased from the marine commercial markets in Yeosu city (Jeonnam, Korea) and retained in glass containers (45.7 × 35.6 × 30.5 cm) supplied with a continuous flow of aerated contaminant-free seawater. Prior to pollutant exposure, the crabs were acclimated in glass containers at 19 ± 1 °C temperature, 25 % salinity, and 12 h light-dark period for 1 d. The crabs were fed with a small amount (~200 mg) of TetraMin (Tetra-Werke, Melle, Germany) daily. All experimental processes were conducted based on the guidelines of the Chonnam National University Institutional Animal Care and Use Committee. The date of approval for the animal experiment was 11 October 2019 (ethical code: CNUIACUC-YS-2019-6C).

### 2.2. Exposure Experiments

PFOS, irgarol, and BPA (99.9% pure) were purchased from Sigma-Aldrich (St. Louis, MO, USA). DEHP solutions were prepared from a solid compound (99%, Junsei Chemical Co. Ltd., Tokyo, Japan). The stock solutions were prepared as 10 mg L^−1^ each of PFOS, irgarol, DEHP, and BPA dissolved in 99% acetone at room temperature. The solvent control concentration was <0.5% acetone. For sublethal exposure, 10 crabs were treated with one of three concentrations of each chemical (1, 10, and 30 µg L^−1^) for an exposure period of 24 h, 96 h, and 7 d, based on the previous results [32,33,34,35]. All experiments were performed in triplicate (three different pools of individuals) using independent samples (total of 120 crabs for the study). Each exposure container of seawater was changed daily, adding the equivalent dose of each chemical for experiment periods.

### 2.3. M. japonicus p38 MAPK Gene Identification and Bioinformatic Analysis

The partial sequence of *p38 MAPK* gene was obtained using the 454 GS FLX transcriptomic database of *M. japonicus* crab [36]. The similarities of *p38 MAPK* with other *p38 MAPKs* in crustacean species were analyzed by the NCBI BLAST program (http://www.ncbi.nlm.nih.gov/Blast.cgi). The ExPASy translation program deduced the 199 amino acid (aa) sequence of *M. japonicus p38 MAPK*. The prediction of putative domains was made with the Smart program (http://smart.embl-heidelberg.de/). Multiple sequence alignments were obtained using the Clustal W2 program and displayed using the GeneDoc Program (v2.6.001). The best-fit model of amino acid replacement was determined by the ProtTest program (v4.1.5). Prior to phylogenetic analysis, we used the Gblocks program (v0.91b) (http://phylogeny.lirmm.fr/phylo_cgi/one_task.cgi?task_type=gblocks) to select conserved blocks from multiple alignments. The phylogenic tree was constructed using the Mega X software (v10.04) based on the 27 deduced amino sequences (151 aa) of *p38 MAPK*-related genes in crustaceans and insects. Bootstrap sampling was reiterated 1000 times.

### 2.4. Tissue Distribution and Expression Analysis of p38 MAPK

Total RNA was extracted from *M. japonicus* tissues using the Trizol reagent (Invitrogen, Life Technologies, Carlsbad, CA, USA) according to the manufacturer’s instructions. To investigate the tissue distribution level of *p38 MAPK*, we extracted total RNA from six different tissues including gills, hepatopancreas, muscle, gonad, heart, and stomach. After recombinant DNase I (RNase-free; Takara, Tokyo, Japan) treatment, the concentration and integrity of the RNA were checked by Nano-Drop 1000 (Thermo Fisher Scientific, Carlsbad, CA, USA) and gel electrophoresis. cDNA was synthesized using 2 µg of RNA according to the SuperScript™III RT kit (Invitrogen) protocol. The synthesized cDNA was diluted as 50-fold and kept at −80 °C.

Quantitative real-time PCR (RT-qPCR) was performed in a volume of 20 µL containing 5 µL of diluted cDNA template, 0.5 µL of each primer (10 µM), and 14.0 µL RNase-free water with AccuPower^®^ PCR premix (Bioneer, Daejeon, Korea) using Exicycler^TM^ 96 (Bioneer). The primer sequences were: *p38 MAPK* forward, 5′-TGTTCAGAGGAGCCAATCC T-3′; *p38 MAPK* reverse, 5′-GCTCCATGTCCTCAAAGCTC-3′; *GAPDH* forward, 5′-TGCTGATGCACCCATG TTTG-3′; *GAPDH* reverse, 5′-AGGCCCTGGACAATCTCAAAG-3′. The PCR product size was 177 bp for the *p38 MAPK* gene and 147 bp for the *GAPDH* gene used as an internal reference. The RT-qPCR cycle was programmed as follows: 95 °C for 60 s, followed by 40 cycles of 95 °C for 10 s, 55 °C for 40 s, and 72 °C for 50 s. The RT-PCR baseline was verified using the Exicycler^TM^ 96 real-time system program (v3.54.8). The relative expression levels of *p38 MAPK* were calculated according to the 2^-ΔΔct^ method [37].

### 2.5. Data Analysis

A statistical analysis was conducted using SigmaStat 3.1 (Systat). Data were presented as the mean ± SD. Independent sample *t*-tests were used to analyze significant differences in *p38 MAPK* mRNA expression between the hepatopancreas and gills. Two-way analysis of variance (ANOVA) was conducted to identify the effects of exposure time and each chemical treatment on *p38 MAPK* transcript expression. Significant difference was presented at * *p* < 0.05 and ** *p* < 0.01.

## 3. Results

### 3.1. Sequence Characterization of the M. japonicus p38 MAPK Gene

The partial cDNA sequence of the *M. japonicus p38 MAPK* gene was obtained from the GS-FLX transcriptome database of *M. japonicus* crab. The *M. japonicus p38 MAPK* sequence was 597 bp long, including an ORF of 199 amino acids (Figure 1). The *M. japonicus p38 MAPK* protein contained a functional domain (STKc) that was the mark of *p38 MAPKs*. The functional domain contained the ED site (Glu and Asp), the putative dual phosphorylation motif Thr-Gly-Tyr (TGY), and the substrate-binding site Ala-Thr-Arg-Trp (ATRW). The *M. japonicus p38 MAPK* nucleotide sequence was 96% and 90% homologous with that of *Eriocheir sinensis* (KF582665) and *Scylla paramamosain* (MH341515), respectively. The deduced amino acid sequences of the *p38 MAPK* were 99%, 98%, 92%, and 88% homologous with those of *E. sinensis* (Chinese mitten crab; AGK89796), *S. paramamosain* (green mud crab; AHH29322), *Penaeus vannamei* (Pacific white shrimp; AFL70597), and *Palaemon carinicauda* (ASU54245), respectively. As a result, the sequence of *M. japonicus p38 MAPK* gene showed high homology with those of the other crustacean species (Figure 1).

### 3.2. Phylogenetic Relationship of M. japonicus p38 MAPK with Other p38 MAPK Genes

The phylogenetic analysis demonstrated that the amino acid sequence of *M. japonicus p38 MAPK* shared high similarity (>90%) with other known *p38 MAPK* homologs of crabs (Figure 2). In the crustacean species, *M. japonicus p38 MAPK* gene formed a cluster with one clade including *p38 MAPK* or *MAPK14* genes from *E. sinensis*, *S. paramamosain*, *P. vannamei*, and *Penaeus japonicus*. Another clade was composed of *p38 MAPK* homolog genes from crustacean species, including various prawns and shrimps. In insect species, *p38 MAPK* homolog genes from mosquitoes were phylogenetically closer to *M. japonicus p38 MAPK* gene than other insects (Figure 2). Insect *p38 MAPK* homologous genes formed one clade with the *p38 MAPK* of the cat flea (*Ctenocephalides felis*), silkworm (*Bombyx mori*), turnip sawfly (*Athalia rosae*), and other related insect species (Figure 2).

### 3.3. Tissue Distribution Levels of M. japonicus p38 MAPK

To detect the tissue distribution of *p38 MAPK*, the expression of *p38 MAPK* transcripts was analyzed in six tissues (hepatopancreas, gills, gonad, muscle, stomach, and heart) of a healthy *M. japonicus* mud crab (Figure 3). The relative expression levels of *p38 MAPK* mRNA were detected in six tissues, with the highest level in hepatopancreas, followed by gills, gonad, and stomach. Relatively low levels of *M. japonicus p38 MAPK* expression were observed in the muscle and heart.

### 3.4. Expression Profiles of p38 MAPK after PFOS or Irgarol Exposures

As shown in Figure 4, a remarkable increase in *p38 MAPK* gene expression was noted in the gills and hepatopancreas of *M. japonicus* crab exposed to PFOS or irgarol. The transcriptional levels of *p38 MAPK* were significantly upregulated in the gills at 30 µg L^−1^ PFOS compared to that in the control for 4 and 7 days (*p* < 0.05) (Figure 4A). In the hepatopancreas, a significant (*p <* 0.05) increase in the expression of *p38 MAPK* was found at most concentrations of PFOS for all exposure times (Table 1). The highest expression of *p38 MAPK* showed a relatively high concentration of 30 µg L^−1^ PFOS at day 4 (*p <* 0.01). In 30 µg L^−1^ PFOS exposure, *p38 MAPK* expression was significantly different among exposure times.

The *p38 MAPK* mRNA expression in the gills significantly increased at day 7 after irgarol exposures (*p <* 0.05) (Figure 4C). In the gills, the late response of *p38 MAPK* to irgarol exposure showed highly significant differences at 1 (3.3-fold) and 30 µg L^−1^ (3.9-fold) compared with control (*p <* 0.01). In contrast to the expression in the gills, transcriptional expression of *p38 MAPK* was induced in the hepatopancreas tissue of *M. japonicus* crab after 24 h of irgarol exposure stress (Figure 4D). For all exposure times, a significant increase in *p38 MAPK* was found at all concentrations of irgarol except at 30 µg L^−1^ on day 4 (*p* < 0.05). On day 7, the *p38 MAPK* transcript expression was significantly upregulated at 1 (3.4-fold), 10 (3.9-fold), and 30 µg L^−1^ (4.8-fold) (*p* < 0.01) irgarol exposures in a concentration-dependent manner (Table 1). Moreover, the *p38 MAPK* expression pattern to irgarol exposure was significantly different among exposure times in the hepatopancreas. With exposures to different PFOS and irgarol concentrations, the *p38 MAPK* expression levels were significantly different between the gills and hepatopancreas (*p* < 0.001).

### 3.5. Expression Profiles of p38 MAPK after DEHP or BPA Exposures

The expression profiles of *M. japonicus p38 MAPK* in the gills and hepatopancreas after DEHP or BPA stress are shown in Figure 5. At day 1, there was no significant difference in *p38 MAPK* expression between control and treated groups in gills to exposures of DEHP and BPA (Figure 5A,C). However, *p38 MAPK* expression significantly increased in hepatopancreas after 1 day of DEHP exposure in a concentration-dependent manner (*p <* 0.05) (Figure 5B). A significant increase in *p38 MAPK* mRNA was found in hepatopancreas at all concentrations of DEHP for all exposure times (*p <* 0.05), whereas an increase in *p38 MAPK* expression was significantly observed in gills at 4 and 7 days after DEHP stress (Table 1). Under DEHP stress, the highest expression of *p38 MAPK* gene was found at 30 µg L^−1^ (4.1-fold) at day 7 in gills and 10 µg L^−1^ (4.3-fold) at day 4 in hepatopancreas (*p <* 0.01). The *p38 MAPK* expression was significantly different among exposure times at 30 µg L^−1^ in gills and 10 µg L^−1^ in hepatopancreas.

For BPA exposure, the mRNA expression of *p38 MAPK* decreased in gills after 1 day of BPA stress (Figure 5C). On day 4, a significant decrease of *p38 MAPK* expression was found in gills at 10 and 30 µg L^−1^ BPA exposures (*p <* 0.05). However, on day 7, *p38 MAPK* mRNA expression significantly increased at all concentrations of BPA (*p <* 0.05) (Table 1). The *p38 MAPK* expression pattern was significantly different in gills between day 7 and other exposure times. In the hepatopancreas, BPA exposure enhanced the expression levels of *p38 MAPK* gene for days 4 and 7 (Figure 5D). A significant increase of *p38 MAPK* expression was found at 10 µg L^−1^ on day 1 and at all concentrations on day 4 in a concentration-dependent manner (*p <* 0.05) (Figure 5D). In the hepatopancreas on day 7, the expression of *p38 MAPK* gene significantly increased at 1 (2.6-fold), 10 (4.2-fold), and 30 µg L^−1^ (3.8-fold) BPA exposures (*p* < 0.05) (Table 1). Moreover, *p38 MAPK* mRNA expression in the hepatopancreas was significantly different among exposure times at 10 µg L^−1^ BPA (*p* < 0.01) (Figure 5D). With exposures to different DEHP and BPA concentrations, the *p38 MAPK* transcript levels were significantly different between the gills and hepatopancreas (*p* < 0.005).

## 4. Discussion

In the present study, a *p38 MAPK* containing a functional domain (STKc) was sequenced from the intertidal mud crab *M. japonicus*. The ORF of *M. japonicus p38 MAPK* gene included the ED site, TGY motif, and substrate-binding site (ATRW) that were highly conserved as reported in all *p38* subfamily members [1,15,38]. *p38 MAPK* activation was triggered by the dual phosphorylation of both Thr and Tyr in the TGY motif, and the ATRW domain as the kinase interaction motif (KIM) docking site is required for binding to the linear KIM sequences and MAPK phosphatases [16,39]. In addition, the ED site is critical for the interaction of *p38 MAPK* with substrates, regulators, and activators [40]. All these studies suggested that *p38 MAPK* in *M. japonicus* contained functions and regulatory mechanisms similar to *p38 MAPK* families in other species.

The basal level of *p38 MAPK* gene is generally expressed in multiple tissues including the gill, liver, brain, kidney, and spleen [1,41]. In the present study, the finding revealed that expression of *p38 MAPK* transcripts was observed on all the tested organs. The highest level of *p38 MAPK* mRNA was found in the hepatopancreas and gills. In crustaceans, the hepatopancreas, a homologous organ to the liver, plays a key role in immunity and metabolic process [42,43]. The crustacean gill has pivotal functions in osmotic regulation and ionic exchanges as well as respiration and is important for the inactivation or degradation process as an immune defense against bacterial or microbial infections [44,45,46,47]. The *p38 MAPK* expression was mostly found in gills and hepatopancreas as well as hemocytes as the innate immune organ in *S. paramamosain* crab [15]. These results indicated that the transcriptional responses of *p38 MAPK* gene could be related to the immune protection of crustaceans.

The aquatic environment has been threatened continuously by exposure to environmental pollutions from agricultural run-off and the abuse of industrial products or compounds. A ubiquitous environmental pollutant, PFOS, is a typical persistent organic pollutant because of resistance to chemical and biological degradation and produces multiple ranges of toxic effects including neurotoxicity [24,48]. In the present study, different level exposures to PFOS increased the transcriptional expression of *M. japonicus p38 MAPK*. The response of *p38 MAPK* was generally higher in the hepatopancreas than in the gill tissue, except for the relative high concentration of PFOS at day 7. The activation of *p38 MAPK* signals due to PFOS was found in male reproductive disorder and was associated with disruption of the blood–testis barrier [24]. PFOS-induced oxidative stress enhanced the *p38 MAPK* pathway [49]. PFOS exposure also leads to endocrine and metabolic dysfunction in juvenile *Cynoglossus semilaevis* through the alteration of expression levels of hormones including insulin, glucagon, and somatostatin [1].

The economic development of the aquaculture industry has been requiring the use of more disinfectants and antifoulant compounds (biocides) for eliminating the microorganisms in aquaculture facilities. Biocides are chemical substances that can kill or prohibit microorganisms responsible for biofouling. In the marine environment, the use of biocides such as irgarol 1051 has proved to be harmful to algae and aquatic organisms [4,50]. In the present study, expression levels of *p38 MAPK* were significantly upregulated in *M. japonicus* hepatopancreas after short- and long-term exposure to different irgarol concentrations, whereas a significant level was found only in gills for long-term exposure of irgarol. The result was the first to be reported about the potential effects of the immune defense involving crustacean *p38 MAPK* pathway to exposure of irgarol. In addition, evidence suggests that irgarol induces disruption of the endocrine system, cellular homeostasis, and exoskeleton chitin formation in marine organisms [33,51,52].

DEHP, a widespread environmental pollutant, can be released from the substrates into the aquatic environment, although it is not covalently bound to the plastic matrix [53]. DEHP-induced hepatic enzymes contribute to homeostasis perturbation of the thyroid hormone, which plays important roles in physiological processes, such as energy metabolism, differentiation, and development [53,54]. DEHP altered the transcriptional expression of p53-regulated apoptosis pathways and suppressed mRNA levels of antioxidant genes. In addition, neurotoxicity to DEHP exposure was found in medaka fish through alteration of the growth and locomotion [55]. In the present study, an upregulation of the *p38 MAPK* mRNA was found in *M. japonicus* gills and hepatopancreas exposed to different doses of DEHP. The increase in *p38 MAPK* was significant at all concentrations and exposure times in the hepatopancreas tissue after DEHP exposure. DEHP induced cancer cell proliferation by the regulation of cell cycle-related genes by increasing p38 expression [20]. Immunosuppression was also caused by DEHP exposure by regulating the expression of inflammatory cytokines and cytochrome P450 homeostasis [8].

BPA is discharged into rivers and marine environments from the migration of BPA-based products or effluents [56]. BPA has caused a significant increase in intracellular ROS levels, reproductive toxicity, induction of apoptosis-related gene expression, damage to mitochondria, and alteration of cellular development because of its endocrine disruptive effects [57,58,59]. In the present study, BPA significantly increased the expression of *p38 MAPK* transcripts. The upregulation of *p38 MAPK* was found at all concentrations of BPA for all exposure times, whereas a significant increase in *p38 MAPK* was found for only long-term exposure of BPA. These observations were in agreement with a previous study, which reported that vitellogenin mRNA expression increased significantly in mud crab exposed to BPA [52]. In addition, BPA induced apoptosis by the activation of *p38 MAPK* [57].

The *p38 MAPK* pathway regulates multiple cellular processes including proliferation, differentiation, apoptosis, and inflammatory responses [1,15,59]. Overexpression of *p38 MAPK* triggered the inhibition of viral gene transcription and protein synthesis as cellular immune responses [60]. Altered responses of *p38 MAPK* gene have been related to stressors including ammonia, hypoxia, salinity changes, and bacterial infections [1,16,23]. The chemical environmental pollutants also induced the upregulation of the *M. japonicus p38 MAPK* transcript in the present study. These results indicated that the *p38 MAPK* can play a key role in the immune defense process against oxidative stress induced by environmental pollutions such as PFOS, irgarol, DEHP, and BPA. The activation of the *p38 MAPK* transcript could be considered as a cellular defense function to anti-stress responses of *M. japonicus p38 MAPK* against exposure toxicities of pollutants. The different expression profiles of *p38 MAPK* suggested that toxic pollutants may trigger the same or different mechanisms including inflammation, apoptosis, and cell cycle regulation through the *p38 MAPK* signaling pathway.

## Figures and Tables

**Figure 1 genes-11-00958-f001:**
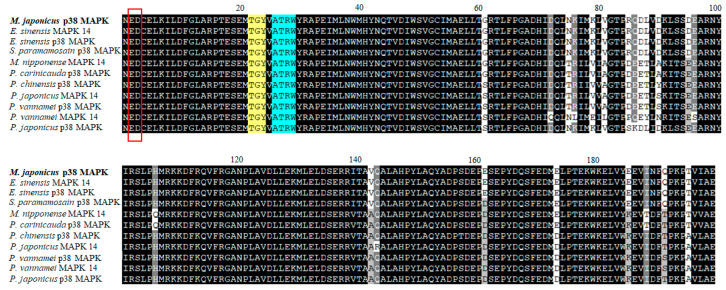
Multiple-sequence alignment of the deduced *Macrophthalmus japonicus* p38 mitogen-activated protein kinase *(MAPK*) gene sequences with the homologous sequences of other crustaceans. Completely conserved residues across all species aligned are shaded in black. The predicted phosphorylation motif TGY (yellow) and the substrate-binding site ATRW (blue) are indicated by color-shaded boxes. The ED site is presented as box.

**Figure 2 genes-11-00958-f002:**
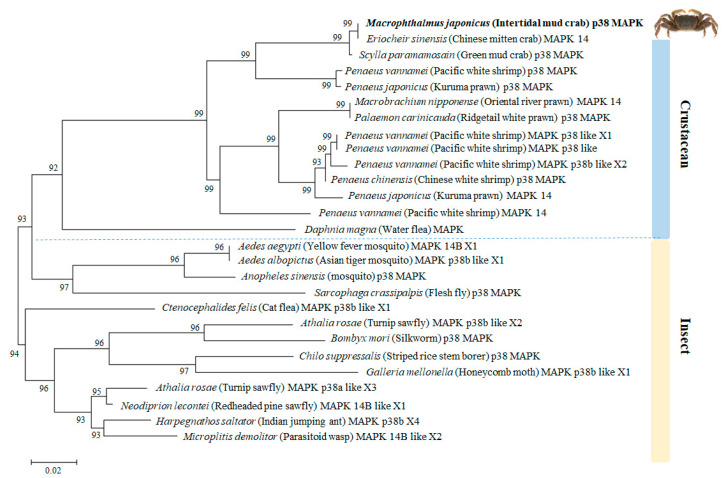
Phylogenetic relationship of *M. japonicus p38 MAPK* gene with other reported *p38 MAPKs*. The phylogenetic tree of the alignment amino acid sequences was constructed by neighbor-joining analysis using the MEGA 4.0 program. Bootstrap values (1000 replicates) are indicated at the nodes. The bar indicates the genetic distances (0.02). GenBank accession numbers for *p38 MAPKs* are: *M. japonicus p38 MAPK* (MT811543), *Eriocheir sinensis* (AGK89796), *Scylla paramamosain* (AHH29322), *Penaeus vannamei* (AFL70597), *Penaeus japonicus* (ANA91276), *Macrobrachium nipponense* (ASM46958), *Palaemon carinicauda* (ASU54245), *Penaeus vannamei MAPK* p38 like X1 (XP_027223540), *Penaeus vannamei MAPK* p38 like (AHE40497), *Penaeus vannamei MAPK* p38 like X2 (XP_027223541), *Penaeus chinensis* (AIY23112), *Penaeus japonicus MAPK14* (BAK78916), *Penaeus vannamei MAPK14* (AGC92010), *Daphnia magna* (KZS17293), *Aedes aegypti* (XP_001653240), *Aedes albopictus* (XP_029724310), *Anopheles sinensis* (KFB51279), *Sarcophaga crassipalpis* (BAF75366), *Ctenocephalides felis* (XP_026462668), *Athalia rosae* (XP_012263336), *Bombyx mori* (NP_001036996), *Chilo suppressalis* (ANO46359), *Galleria mellonella* (XP_026758639), *Athalia rosae* (XP_012263336), *Neodiprion lecontei* (XP_015509390), *Harpegnathos saltator* (XP_011137910), *Microplitis demolitor* (XP_008557345).

**Figure 3 genes-11-00958-f003:**
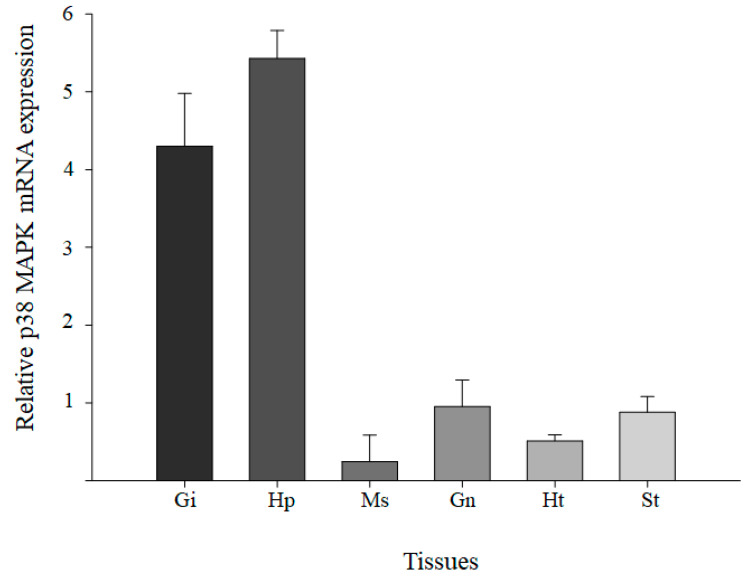
The distribution of *M. japonicus p38 MAPK* transcripts in different tissues (Gi, gill; Hp, hepatopancreas; Ms, muscle; Gn, gonad; Ht, heart; St, stomach). Each tissue was collected from 10 mud crabs. The experiments were performed in triplicate. The relative expression levels of each gene in each tissue were normalized by GAPDH transcript and data were presented as the mean ± SD.

**Figure 4 genes-11-00958-f004:**
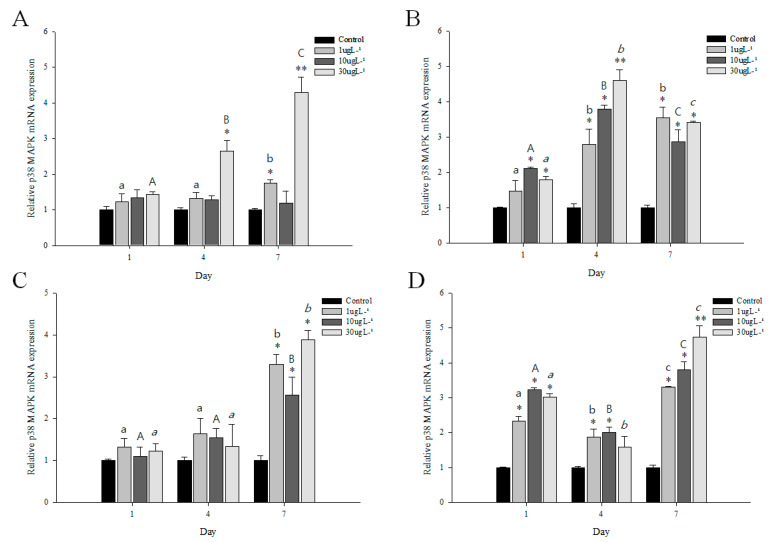
Relative expression of *p38 MAPK* mRNA in *M*. *japonicus* gills (**A**,**C**) and hepatopancreas (**B**,**D**) exposed to 1, 10, and 30 μg L^−1^ perfluorooctane sulfonate (PFOS; **A**,**B**) and irgarol (**C**,**D**) at days 1, 4, and 7. The values were normalized against GAPDH. Values of each bar represent the mean ± SD. A statistically significant difference is presented by an asterisk at * *p* < 0.05 and ** *p* < 0.01 compared with the control (relative control value of *p38 MAPK* = 1).

**Figure 5 genes-11-00958-f005:**
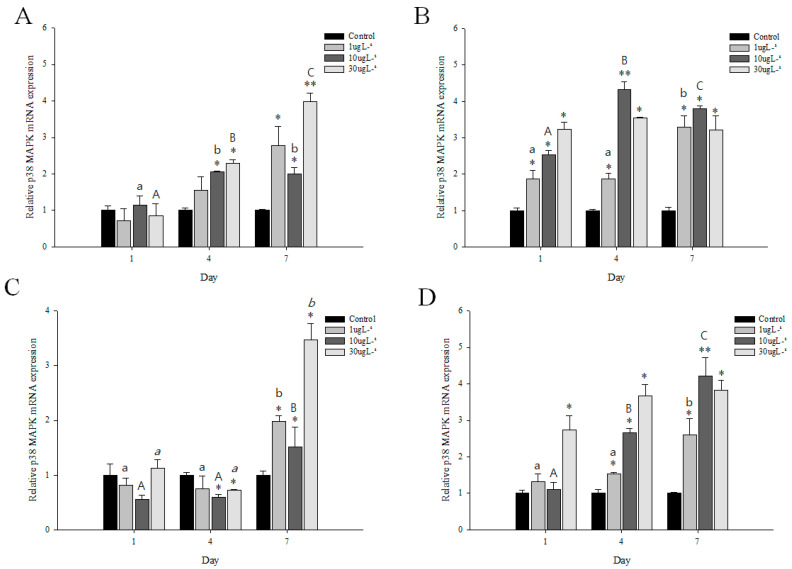
Relative expression of *p38 MAPK* mRNA in *M*. *japonicus* gills (**A**,**C**) and hepatopancreas (**B**,**D**) exposed to 1, 10, and 30 μg L^−1^ DEHP (**A**,**B**) and BPA (**C**,**D**) at days 1, 4, and 7. The values were normalized against GAPDH. Values of each bar represent the mean ± SD. A statistically significant difference is presented by an asterisk at * *p* < 0.05 and ** *p* < 0.01 compare with the control (relative control value of *p38 MAPK* = 1).

**Table 1 genes-11-00958-t001:** Summary of relative expression level of *p38 MAPK* in gills and hepatopancreas of *M. japonicus* that had been exposed to different doses of 1, 10, and 30 μg L^−1^ PFOS, irgarol, di(2-ethylhexyl) phthalate (DEHP), bisphenol A (BPA), and solvent control at different time points (days 1, 4, and 7) by a color schematic representation (green color: low expression level of *p38 MAPK* compared with the control, red color: high expression level of *p38 MAPK* compared with the control). Each transcript was determined using GAPDH by the 2^−^^ΔΔCt^ method. Relative fold-change in gene expression was determined by dividing the average relative expression of each individual at each time point by the average relative expression of solvent control individuals at day 1 (*n* = 120). Relative fold change mRNA expression for *p38 MAPK* gene for control at day 1 = 1.

		Gill				Hepatopancreas				
		PFOS	Irgarol	DEHP	BPA	PFOS	Irgarol	DEHP	BPA	
Control		1.00	1.00	1.00	1.00	1.00	1.00	1.00	1.00	
1 µg L^−1^	Day1	1.23	1.32	0.71	0.82	1.47	2.34	1.87	1.32	
	Day4	1.33	1.64	1.56	0.75	2.80	1.88	1.88	1.54	
	Day7	1.76	3.30	2.78	1.98	3.54	3.30	3.30	2.61	
10 µg L^−1^	Day1	1.35	1.10	1.14	0.50	2.11	3.23	2.54	1.10	
	Day4	1.28	1.54	2.05	0.60	3.78	2.00	4.32	2.67	0.2–0.5
	Day7	1.20	2.56	2.00	1.52	2.87	3.80	3.80	4.21	0.51–0.99
										1.0
30 µg L^−1^	Day1	1.43	1.22	0.86	1.12	1.80	3.01	3.23	2.73	1.01–1.5
	Day4	2.66	1.34	2.30	0.72	4.60	1.59	3.54	3.67	1.51–4.0
	Day7	4.30	3.88	4.10	3.47	3.42	4.73	3.22	3.82	>4.0

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
