# Peer review of "Expression Levels of the Immune-Related p38 Mitogen-Activated Protein Kinase Transcript in Response to Environmental Pollutants on Macrophthalmus japonicus Crab"

_genes, 2020, doi:10.3390/genes11090958_

Round 1

Reviewer 1 Report

In this manuscript, the authors describe the immune-related p38 mitogen-activated protein kinase (p38 MAPK) transcript in the Macrophthalmus japonicus crab. After a bioinformatics to identify and characterize the p38 MAPK, the authors used a RT-qPCR approach to analyse the gene expression level of this marker after exposure to classical environmental pollutants.
This is an interesting and valuable study to increase our understanding of the response of an economically important clade (crustaceans) to environmental pollutants.

The manuscript is well written and organized, the reviewers has several minor concerns, mainly concerning methodological precisions to add.

The phylogenetic tree including 27 sequences of p38 MAPK is based on an amino acid alignment, and constructed by the NJ approach (only indicated in the legend of the Figure 2, not on the M&M). Could you specify the final alignment length used to build this tree? (I did not find this information in the manuscript). Did you use an approach like Gblocks to remove some ambiguously aligned part of your dataset (see Talavera & Castresana (2007): "Improvement of phylogenies after removing divergent and ambiguously aligned blocks from protein sequence alignments" in Systematic Biology)? Furthermore, it is also better to specify the best-fit models of amino acid replacement applied to your data (see Prottest software to proceed, from Darriba et al., 2011).

Concerning the p38 MAPK expression analysis performed, it is unclear to me when you say that each value indicated in the Figure 3, 4 and 5 correspond to the mean expression of 3 different pools of individuals (biological triplicate) or 3 different measure from the same pool of individuals (technical triplicate)... Please precise it in the final manuscript.

To complete this study, it might be valuable to test this RT-qPCR on crabs originated from natural populations... I guess this is probably already planned (or even performed).

Author Response

Reviewer1

Comments and Suggestions for Authors

In this manuscript, the authors describe the immune-related p38 mitogen-activated protein kinase (p38 MAPK) transcript in the Macrophthalmus japonicus crab. After a bioinformatics to identify and characterize the p38 MAPK, the authors used a RT-qPCR approach to analyse the gene expression level of this marker after exposure to classical environmental pollutants.

This is an interesting and valuable study to increase our understanding of the response of an economically important clade (crustaceans) to environmental pollutants.The manuscript is well written and organized, the reviewers has several minor concerns, mainly concerning methodological precisions to add.

The phylogenetic tree including 27 sequences of p38 MAPK is based on an amino acid alignment, and constructed by the NJ approach (only indicated in the legend of the Figure 2, not on the M&M).

  • We added the detailed Material and methods about the phylogenetic analysis in line 102 and lines 109~110 of the revised manuscript.

Could you specify the final alignment length used to build this tree? (I did not find this information in the manuscript).

  • The final alignment length to build phylogenetic tree was 155 aa (27 species) of total deduced 199 aa sequence of M. japonicus p38 MAPK after analysis using prottest and Gblocks programs. We added the final alignment length in line 109 of the revised manuscript.

Did you use an approach like Gblocks to remove some ambiguously aligned part of your dataset (see Talavera & Castresana (2007): "Improvement of phylogenies after removing divergent and ambiguously aligned blocks from protein sequence alignments" in Systematic Biology)?

  • We used Gblock program to remove some ambiguously aligned part of the multiple alignment. We added the material and methods as “Prior to phylogenetic analysis, we performed Gblocks program (ver. 0.91b) (http://phylogeny.lirmm.fr/phylo_cgi/one_task.cgi?task_type=gblocks) to select conserved blocks from multiple alignments.” in lines 106~108 of the revised manuscript.

Furthermore, it is also better to specify the best-fit models of amino acid replacement applied to your data (see Prottest software to proceed, from Darriba et al., 2011).

  • We also used prottest program (ver. 4.1.5) to determine best-fit models of amino acid replacement in lines 105~106 of the revised manuscript.

Concerning the p38 MAPK expression analysis performed, it is unclear to me when you say that each value indicated in the Figure 3, 4 and 5 correspond to the mean expression of 3 different pools of individuals (biological triplicate) or 3 different measure from the same pool of individuals (technical triplicate)... Please precise it in the final manuscript.

  • We measured the mean expression of the M. japonicus p38 MAPK in three different pools of individuals (biological triplicate) as line 95 of the revised manuscript.

To complete this study, it might be valuable to test this RT-qPCR on crabs originated from natural populations... I guess this is probably already planned (or even performed).

  • Further study, we will plan to test this RT-qPCR on crabs originated from natural population as the reviewer’s suggestion.

Reviewer 2 Report

I think that you are exploring an important area to clarify the influence of the polluted environment on the physiological function of crustaceans, however in my opinion some aspects require further explanation in your work, I have some questions and comments that I hope will contribute to the exposition of your study .

You used 4 different chemical contaminants and apparently they act on the same cascade that regulates the expression of p38 MAPK. The question is, do all the used pollutants have the same mechanism of action to validate their effect on a common factor?

line 2: I think it is convenient to change "molecular responses" because only the increase in expression level was obtained.

lines 28-29-30: I suggest changing the final conclusion or explaining the identified anti-stress properties.

lines 69 y 70: Is the detection of human cancer cells using lectin related with p38 MAPK?

line 76-77: please review this idea.

line 81: Are the animals used in the experiments grown in contaminant-free water?

line 92: What criteria did they use to determine the experimental concentration of the compounds used? because the animals could be adapted to much higher levels of pollutants in their natural environment

line 93: Reference 27 analyzed the effect of heavy metals. It is not appropriate to cite it as an antecedent in the concentration of tested compounds. Add references for Irgarol and DEHP

line 95: Isn´t lack of food considered a stressor in this species?

line 176: I suggest reporting the identified sequence to the GenBank database, in les than two days they will send you the access number

line 192: capital letter is generally used (P < 0.05)

line 219: It is not possible to make valid conclusions with such a small sample (n = 3) for this type of study, is recommended an n=24, at least. You could reevaluate in the most representative concentrations and times of your study.

line 280: How do you interpret the lower expression of p38 MAPK in the hepatopancreas after 7 days of PFOS exposure?

line 285: specify the species from the cited article.

lines 328-329: In my opinion, the final conclusion can’t be supported by the results of this study.

line 202: irga-ol

line 80: a space is missing

title of the Table 1:  correct the word "Heaptopancreas"

Author Response

Reviewer2

Comments and Suggestions for Authors

I think that you are exploring an important area to clarify the influence of the polluted environment on the physiological function of crustaceans, however in my opinion some aspects require further explanation in your work, I have some questions and comments that I hope will contribute to the exposition of your study.

You used 4 different chemical contaminants and apparently they act on the same cascade that regulates the expression of p38 MAPK. The question is, do all the used pollutants have the same mechanism of action to validate their effect on a common factor?

  • Immune-related p38 MAPK plays an important role in various signal processes including the inflammatory response, cell differentiation, cell cycle regulation, and apoptosis. Our results suggested to changed regulation of p38 MAPK mRNA by exposure toxicities of pollutants including PFOS, Irgarol, DEHP, or BPA. These pollutants may induce activation of p38 MAPK-related defense mechanisms to toxic stress through various signal process including inflammatory response, cell differentiation, cell cycle regulation, and apoptosis. Although we can not conclude the same or different mechanism to different pollutants, further study should be need to research about the regulation effects in several genes of MAPK signal pathway depending to each pollutant.
  •  

line 2: I think it is convenient to change "molecular responses" because only the increase in expression level was obtained.

  • We revised as “Expression responses” in line 2 of the revised manuscript.

lines 28-29-30: I suggest changing the final conclusion or explaining the identified anti-stress properties.

  • We revised the final conclusion in the abstract as “The findings in this study supported the anti-stress responses of M. japonicus p38 MAPK was reflected by expressional changes of p38 MAPK signaling regulation as a cellular defense against exposure to environmental pollutants.” In lines 28~30 of the revised manuscript.

lines 69 y 70: Is the detection of human cancer cells using lectin related with p38 MAPK?

  • We revised the sentence as “M. japonicus can be used for the characterization and purification of lectin as well as resources of fishing and fishery” in lines 69-70 of the revised manuscript.

line 76-77: please review this idea.

  • We revised the sentence as “The results suggested potential activation in the immune responses of M. japonicus p38 MAPKs to exposure toxicity as a defense against exposure to environmental pollutants.” in lines 76~77 of the revised manuscript.
  •  

line 81: Are the animals used in the experiments grown in contaminant-free water?

  • Line 82: After purchase of crabs collected from contaminant-free environment, the crab retained in glass containers with a continuous flow of aerated contaminant-free seawater for acclimation to laboratory conditions before the exposure experiments.

line 92: What criteria did they use to determine the experimental concentration of the compounds used? because the animals could be adapted to much higher levels of pollutants in their natural environment.

  • We tested in M. japonicus crab with the experimental concentrations of each compounds used in the previous studies (Park et al., 2016 for irgarol, Kim et al., 2014 and Park et al., 2015 for PFOS, Park et al., 2019 for DEHP and BPA). These concentrations were selected from reference studies with relative levels found in field water or sediment environment.
  •  
  • Irgarol : 14~ 22 ng L-1 in stream and small river (Luft et al., 2014) and up to 67.64 ng L-1 in Korean coasts (Lee et al., 2010).
  • PFOS: 0.35~47 ng L-1 in the west coast (Naile et al., 2013) and 2.7 μg L-1 in the water of the Anoia river in Spain (Campo et al. 2015).
  • BPA: 4~21 ng L-1 in U.S. river estuaries (Kolpin et al., 2002; Dong et al., 2009).
  • DEHP: 0.47~12 µg L-1 in the china river estuary (Li et al., 2016).

line 93: Reference 27 analyzed the effect of heavy metals. It is not appropriate to cite it as an antecedent in the concentration of tested compounds. Add references for Irgarol and DEHP

  • We revised the reference with deletion of 27 and addition of 34 in line 94 of the revised manuscript (Reference 31 for PFOS, Ref. 32 for irgarol, Ref. 33 for BPA, added Ref. 34 for DEHP).

line 95: Isn´t lack of food considered a stressor in this species?

  • We revised the incorrect sentence as “The crabs were fed with a small amount (~ 200 mg) of Tetramin (Tetra-Werke, Melle, Germany) daily.” in lines 84~85 of the revised manuscript.

line 176: I suggest reporting the identified sequence to the GenBank database, in les than two days they will send you the access number

  • We added the GenBank access number of M. japonicus p38 MAPK sequence as “MT811543” in line 170 of the revised manuscript.

line 192: capital letter is generally used (P < 0.05)

  • We revised as capital letter of the P value in the whole manuscript (line195, line198, line 201, line 211, line 245, line 247, line 248, line 250)

line 219: It is not possible to make valid conclusions with such a small sample (n = 3) for this type of study, is recommended an n=24, at least. You could reevaluate in the most representative concentrations and times of your study.

  • There are some error in the line 222~223. It (n=3) is a misspelled number. As the material and method (line 96), the total crabs are 120 individuals for the study. We revised individual number (n=120) used for the experiment in the revised manuscript.  

line 280: How do you interpret the lower expression of p38 MAPK in the hepatopancreas after 7 days of PFOS exposure?

  • We revised the sentence as “ The response of p38 MAPK was generally higher in the hepatopancreas than in the gill tissue, except to the relative high concentration of PFOS at day 7” in lines 283~284 of the revised manuscript.
  •  

line 285: specify the species from the cited article.

  • We added the specific species (juvenile Cynoglossus Semilaevis) in lines 287~289 of the revised manuscript.

lines 328-329: In my opinion, the final conclusion can’t be supported by the results of this study.

  • We revised the final conclusion as “The activation of the p38 MAPK transcript could be considered as a cellular defense function to anti-stress responses of M. japonicus p38 MAPK against exposure toxicities of pollutants. The different expressions of p38 MAPK suggest that toxic pollutants may trigger same or different mechanisms including inflammation, apoptosis, and cell cycle regulation through p38 MAPK signaling pathway” in lines 331~335 of the revised manuscript.

line 202: irga-ol

  • We revised the error from “irgaol” to “irgarol” in line 205 of the revised manuscript.

line 80: a space is missing

  • We added a space in line 80 of the revised manuscript.

title of the Table 1:  correct the word "Heaptopancreas"

  • We revised the error from “Heaptopancreas” to “Hepatopancreas” in line 218 of the revised manuscript.

Round 2

Reviewer 2 Report

I suggest you to review the English language edition mainly in the last sentences of the Abstract, Introduction and Discussion.

Line 29:  I suggest "expression levels" instead of expressional

Line 140: confirm if the reference cited here is correct [35]

title of Table 1:  correct the word "Heaptopancreas" immediately after the line225

Author Response

Comments and suggestions for Authors

I suggest you to review the English language edition mainly in the last sentences of the Abstract, Introduction and Discussion.

--> We revised the English language in the last sentences of the Abstract (lines 28-30), and Introduction and Discussion (lines 75-77).

Lines 28-30: The findings in this study support the anti-stress responses against exposure to environmental pollutants was reflected to changes of expression levels in the M. japonicus p38 MAPK signaling regulation as a cellular defense mechanism.

Lines 75-77: The results suggest potential activation of M. japonicus p38 MAPKs to the immune responses as a defense against exposure toxicities to environmental pollutants.

Line 29: I suggest “expression levels” instead of expressional

--> Lines 29-30: We revised as “expression levels” instead of “expressional” as the reviewer’s comment in the revised manuscript.

Line 140: confirm if the reference cited here is correct [35]

--> Line 139: We deleted the reference [35] in the revised manuscript.

Title of Table: correct the word “Heaptopancreas” immediately after the line225

--> Line 225: we corrected the word “Heaptopancreas” on the Table 1.